

# 3D-hydrodynamic modelling of flood impacts on a building and indoor flooding processes

B. Gems[1], B. Mazzorana[2], T. Hofer[3], M. Sturm[1], R. Gabl[1], M. Aufleger[1]

[1]Unit of Hydraulic Engineering, Institute for Infrastructure Engineering, University of Innsbruck, Innsbruck, 6020, Austria
[2]Institute of Environmental and Evolutive Sciences, Faculty of Sciences, Universidad Austral de Chile, Valdivia, 14101, Chile
[3]MWV Bauingenieure AG, Baden, 5400, Switzerland

*Correspondence to*: B. Gems (bernhard.gems@uibk.ac.at)

**Abstract.** Given the current challenges in flood risk management and vulnerability assessment of buildings exposed to flood hazards, three-dimensional numerical modelling of torrential floods and its interaction with buildings are presented. By means of a case study application, the FLOW-3D software is applied to the lower reach of the Rio Vallarsa torrent in the village of Laives (Italy). A single-family house on the flood plain is thereby considered very detailed and exposed to a 300 yr flood hydrograph. Different building representation scenarios, amongst an entire impervious building envelope and the assumption of fully permeable doors, windows and cellar shafts, are analysed. The modelling results give insight into the flooding process of the building's interior, the impacting hydrodynamic forces on the exterior and interior walls and, further, quantify the impact of flooding a building on the flow field on the adjacent flood plain. The presented study means a step towards the development of a comprehensive physical vulnerability assessment framework. For pure water floods, it shows possibilities and limits of advanced numerical modelling techniques within flood risk management and, thereby, the planning of local structural protection measures.

## 1 Introduction, vulnerability assessment within integral flood risk management

Recently, researchers with different scientific background proposed major contributions to a better understanding of the concept of vulnerability, each according to their specific disciplinary focus (Hufschmidt, 2011; Fuchs, 2009). Traditionally, when addressing vulnerability, social scientists tend to emphasize the characteristics of people or communities in terms of their capacity to anticipate, cope with, resist, and recover from the impact of a hazard (e.g. Wisner, 2004). In contrast, from a purely engineering perspective, vulnerability is defined as the degree of loss to an element at risk as a result of a hazard impact with a given intensity and frequency (Fell et al., 2008). Vulnerability is assessed on the basis of empirical data and/or scenario modelling.

The efforts to increase the resilience of communities towards natural hazards have to be devoted (i) to a substantial reduction of the vulnerability of the built environment (e.g. critical infrastructure, residential buildings) and (ii) to the promotion of management actions with respect to the different possibilities given by the risk management cycle in Fig. 1.



Regarding the respective societal framework, risk assessment is targeted at the evaluation of risk, which includes (i) a social assessment of the level of risk to be accepted, (ii) an economic valuation of possible mitigation activities versus the level of risk reduction achieved by these measures, (iii) a general assessment of individual and societal risk awareness and acceptance as well as (iv) possibilities of risk mitigation. Optimal mitigation strategies seek to address the different and

interrelated dimensions of vulnerability such as physical (structural), social, economic, and institutional vulnerability. Hereby, the reduction of structural or physical vulnerability is seen as a starting point, resulting directly in a reduction of physical losses and indirectly in a mitigation of possible consequences with respect to the other dimensions of vulnerability (Kappes et al., 2012a; Kappes et al., 2012b; Papathoma-Köhle et al., 2011; Fuchs, 2009). This implies that vulnerability assessment should meet high quality standards to provide for an integrated knowledge basis for all relevant management

options, including the design of appropriate mitigation and the policy implementation during necessary decision-making actions.

Amongst others, integral risk management covers structural (technical) measures for protection against natural hazards. Aiming for a reduction of risk, they actively reduce hazard potential. For the case of torrential hazards, afforestation measures, erosion control, check dams and levees are typically applied. If decreasing the damage potential, e.g. in terms of

object protection, technical measures have a passive effect. Basically, hazard analysis by means of experimental and numerical modelling of relevant scenarios became increasingly important in recent past. In case of numerical modelling, significant advances in modelling techniques and the augmented computational power presently allow for analyses of complex issues and scenarios (e.g. Gems et al., 2014a; Gems et al., 2014b; Mazzorana et al., 2014; Chiari, 2008). They enable the simulation of hazard processes on a relatively large spatial scale (e.g. Gems, 2014c; Gems et al., 2012). Hydraulic

scale models are realized mostly to address and model complex problem settings, geometrical configurations and compound scenarios, e.g. morphodynamics (sediment transport) in a complex three-dimensional flow field, flow-structure interaction and involved bed-load transport, impacts of hazard processes on structures (e.g. Scheidl et al., 2013; Armanini and Scotton, 1992). Experimental modelling is restricted to a rather limited spatial scale.

Research efforts in the field of integral flood risk management and, thereby, hazard analysis and modelling (Fig. 1) have

been devoted to study the physical vulnerability of buildings exposed to fluvial hazard processes facing (i) the aim to compute vulnerability functions for use in risk assessment (Totschnig and Fuchs, 2013; Papathoma-Köhle et al., 2012; Fuchs et al., 2007) and (ii) the design of local structural mitigation measures (Holub et al., 2012). Despite these efforts, considerable research questions still remained unanswered: while studies firstly combined empirical loss data with information on process magnitudes and resulted in damage-loss or vulnerability functions, the latter studies were mainly

focused from a practical perspective on the reduction of structural vulnerability of individual buildings. Due to the underlying empiricism of such vulnerability functions, the physics of the damage generating mechanisms remains unveiled, and, as such, the applicability of the empirical approach for planning hazard-proof buildings is rather limited. Mazzorana et al. (2014) identified the following essential requirements for vulnerability assessment of buildings exposed to fluvial hazard processes:





(a) A comprehensive concept of vulnerability evaluation requires a physics-based approach with a detailed representation of the impacting hazard process, both, with respect to space and time.

(b) Quantification of the resulting impacts on a building envelope and detection of possible material intrusion processes requires an analysis of the geometrical structure of the building with respect to the time-varying flow field of the impacting process and, if geo-mechanical actions may interfere, with respect to the residual bearing capacity of the soil layers the object is situated.

(c) A physical response (resistance) analysis of the building structure considering the time-varying impacts is required, both, from a structural analysis perspective (statics, elastostatics and dynamics) and a building physics viewpoint. Stresses and strains on the building have to be compared with maximum admissible values (according to the set of norms EN 1990 (Eurocode 0: Basis of Structural Design), EN 1991 (Eurocode 1: Actions on Structures) and the specific design codes EN 1992 to EN 1999).

Referring to these basic requirements, Mazzorana et al. (2014) defined a five-step-procedure according to Fig. 2 in order to reliably assess the physical vulnerability of elements at risk. The proposed concept is directed at unveiling the sequences of significant loss generation mechanisms, both methodologically and computationally. By evaluating potential damages, the scope of vulnerability assessment is expanded beyond its classical role as a decision-support tool and is closely linked to the planning process of torrent control measures. The workflow requires the definition of a suitable control volume and convenient control sections for each considered element at risk. Process and impact modelling (steps A and B according to Fig. 2) lead to a spatially explicit and time-varying quantification of actions and effects on the building structure. The response model (step C according to Fig. 2) consists in the verification of (i) a set of limit states according EN 1990 (ultimate limit states ULS and serviceability limit states SLS) and (ii) the non-intrusion condition for the liquid and solid material. Details on the steps of damage accounting and economic loss valuation are also covered in Mazzorana et al. (2013; 2012a; 2012b).

Within the context of an analysis of a torrential hazard event and thereby explicitly focusing on the morphodynamic processes and not taking into account any geo-mechanical processes and the building physics, Mazzorana et al. (2014) applied the proposed concept for a residential building located at the fan apex of the Grossberg torrent in the Italian Alps. The study highlighted the circumstance that for medium hazard intensities, vulnerability of buildings critically depends on the patterns of water and material intrusion through openings such as doors, wells and windows. In addition to a proper consideration of the resistance of the considered building in terms of the physical impact and the structural response, also the physical processes taking place on and through the building envelope (e.g. material intrusion and moisture transfer and accumulation, wetting and drying of the outer and inner layers of the building) are found to be relevant within vulnerability assessment.

Due to a lack of data from fundamental research and software limitations, Mazzorana et al. (2014) either not considered specific processes and analytical steps of the assessment scheme (Fig. 2) or analysed them by using empirical data and models, mainly:





(a) The transformation of process parameters (flow velocities, flow depths, bed level changes) to impact parameters (static and dynamic loadings) is based on straightforward empirical approaches estimating the impact of torrent hazards on idealized surfaces.

(b) The processes of water and material intrusion and consequential impacts on the building envelope and on the damage pattern are not considered.

(c) The economic valuation (damage estimation) is based on the application of empirical damage functions relating the loss to the maximum impacting flow depths. However, any dynamics and time-varying process patterns (wetting areas and durations, fluid forces, etc.) have some influence on the impact and response model and thus on the profiles of damage consequences.

(d) The applied case study explicitly considers one specific element at risk. Thus, and due to the non-inclusion of material intrusion processes, any interaction of the relevant elements at risk situated on the Grossberg fan apex has not been analysed. Accordingly, also a geostatistical analysis focusing on the damage patterns and interaction of specific elements at risk situated at different spots on the fan apex has not been accomplished.

In the context of the proposed procedure (based on Mazzorana et al. (2014), the content of the present paper is explicitly addressed on the hydrodynamic simulation of building intrusion processes. A case study analysis is accomplished for a specific element at risk, situated in the sphere of influence of a torrential stream in the Italian Alps. The flow field in the lower reach of the torrent channel, the flood plain in the near surroundings of the considered building and the building's flooding processes are modelled with the FLOW-3D software (Flow Science Inc., 2012), both for a set of steady and unsteady flow conditions. With regard to the aforementioned issues (a) to (c), interaction of the flow fields inside and outside of the building is analysed. Further, impacts on loadbearing walls of the building are evaluated discretely in space and time.

Numerical modelling of flood hazard processes, impacting, entering and flooding a building envelope, has not yet been sufficiently examined with experiments or a numerical model. Therefore the present case study analysis is apriori constrained to pure water floods (WFL according Heiser et al. (2015) and aimed mainly at the following research questions:

- If constraining to pure water floods (WFL) with no involvement of bed-load (Heiser et al. (2015), is there a relevant dynamic impact of the entraining water on the building structure and a noticeable influence on the flow field on the adjacent flood plain?

- With regard to the planning process of local structural protection measures, does the simulation of building flooding processes provide beneficial information for the design and evaluation of protection measures?

- From the perspective of computational capacity and practical application, e.g. for flood plain mapping and hazard zone planning respectively, is it feasible to enlarge the simulation area to a larger extent in order to cover a couple of buildings and objects?



## 2 Case study analysis

### 2.1 Introduction and modelling assumptions

Referring to the aforementioned introduction in vulnerability assessment and the consideration of mutual influences of flood hazard processes and buildings, the work presented within this paper deals with the simulation of building flooding

processes, their influences on the adjacent flow field and the determination of impacting forces on a building structure. In the sense of a case study analysis, focus is put on the flood plain at the Rio Vallarsa in the village of Laives (Autonomous Region of Trentino-Alto Adige, Italy, Fig. 3). One specific element at risk, being distinctively endangered of flooding in case of a torrential hazard event, is considered as a permeable structure within hydrodynamic 3D-numerical modelling (Fig. 5). Therein, solely impacts of pure water hydrographs (WFL according Heiser et al. (2015) are analysed. Any expected

influence of sediments, substantially (i) loss of flow capacity in the torrent channel due to the transport of bed-load (Gems et al., 2014a; Gems et al., 2014b; Hübl et al., 2002; Hunzinger and Zarn, 1996), (ii) intrusion of sediments into the element at risk and (iii) a significant increase of impacting forces compared to clear water conditions (Mazzorana et al., 2014), are not considered. Two basic aspects support the disregard of bed-load transport processes in this specific case:

    (a) Referring to the characterisation of the Rio Vallarsa catchment and the damage causing torrential hazard processes
(Sect. 2.2), a deposition basin including a debris retention dam is located at the alluvial cone closely upstream the case study area (Fig. 3). The basin volume corresponds to the expected amount of sediment during a 150 yr-design event in the catchment. Bed-load is not expected to pass the concrete dam and, thus, the torrent channel downstream is loaded with pure water hydrographs only.

    (b) Modelling of building flooding (discharge) and intrusion (sediment) processes is a topic of current basic research and
not yet explicitly considered within flood risk management, at least in the Alpine space. Reflecting capabilities and limits of numerical models, the simulation of torrential floods with intense sediment loads (WST and DBF according Heiser et al. (2015) is currently restricted to 2D-numerical codes (e.g. Vetsch et al., 2014; Rosatti and Begnudelli, 2013). This contrasts with the requirement of a three-dimensional approach, when the flow field at and the flooding of an element at risk with basement level and first floor, with a complex structure both inside and from the exterior and
with openings (doors and windows, light shafts, etc.) is intended to be numerically modelled. Further, from the perspective of computational demand, applications of 3D-numerical codes are basically limited to rather small river sections and small-scaled areas respectively (Gabl et al., 2014; Gems et al., 2014a; Habersack et al., 2007). Against this background, the presented case study analysis is intended to focus on a specific building and its immediate sphere of influence. The simulations are aimed for use in the context of physics-based vulnerability analysis (Mazzorana et al.,
2014) and the planning of local structural protection rather than large-scale inundation mapping. Thereby, subjects of investigation are also computational effort and limits of numerically modelling the building-fluid-interaction, each from a practical perspective.



## 2.2 Catchment and building characteristics – hazard and damage potential

The Rio Vallarsa catchment is situated to the south of Bolzano in South Tyrol, Italy. Covering 29.4 km² and ranging from 230 m at Laives to 1550 m above sea level, it represents a tributary catchment to the Adige River. The catchment extends mainly in an east-west direction. From a geological perspective, the catchment is shaped by the Bozen quartz porphyry (in.ge.na engineering office, unpublished). In the upper catchment part marginal incisions in glacial deposits and gully erosion characterise the trunk torrent. A straight-line channel with moderate gradients and a few small tributaries can be observed in the middle part of the catchment. Further downstream, the Rio Vallarsa passes a rather narrow and deeply incised canyon before entering the spacious Adige valley at the village of Laives. The torrent passes the settlement area of Laives on the south-west periphery along the border of the valley floor. After leading along agricultural area, the channel enters the Adige River at the village of Ora.

Both, fluviatile and debris flow regimes characterise observed torrential hazard events in the middle and upper catchment and as well the upper section on the alluvial fan (in.ge.na engineering office, unpublished; Fig. 3). Due to a sufficiently dimensioned sediment deposition basin at Laives (Fig. 3), flooding discharges without significant fractions of sediment threaten the settlement and commercial areas further down the deposition basin (in.ge.na engineering office, unpublished).

With regard to the hydrogeological hazard analysis done by in.ge.na engineering office (unpublished), the 100 yr flood peak (HQ100) of the Rio Vallarsa at the village of Laives is approximately 35 m³s⁻¹. The 300 yr flood peak (HQ300) is estimated to 55 m³s⁻¹. The study is based on the common assumption of equal reoccurrence intervals of impacting design precipitation and discharge. The statistical rainfall analysis is thereby based on observed data from the monitoring station at Bronzolo, which is situated three kilometres south of Laives at an altitude of 250 m above sea level. However, a reconstruction analysis of the flood event in November 2012, which basically featured a peak discharge of 55 m³s⁻¹ and bankful conditions in the channel at Laives, indicates higher peaks for the 100 yr- and the 300 yr-event. It was analysed that the observed rainfall intensities and durations in November 2012 featured clearly lower reoccurrence intervals. Consequently, the design flood hydrographs were modified accordingly (Department of Hydraulic Engineering, Autonomous Province of Bolzano, unpublished), amongst leading to a decrease of the 55 m³s⁻¹-reoccurrence interval to about 5 years. Based on these latest data and analyses, sufficient protection against torrential hazards from the Rio Vallarsa catchment is not fulfilled at Laives, since flooding of the rigid torrent channel is already expected for discharges around HQ10 (Department of Hydraulic Engineering, Autonomous Province of Bolzano, unpublished). Buildings and infrastructure in close proximity to the torrent channel are threatened of being flooded in case of a torrential hazard event.

Figure 3 illustrates the situation at the southern part of the alluvial fan at Laives and the track of the Rio Vallarsa. Therein, the case study area is situated straight down the deposition basin. It covers roughly 170 m of the trapezoidal rigid torrent channel, which features a gradient of 1.1 % and a cross section area of 19.5 m². It is a brick work canal lined with cement mortar, whereby the channel side walls are partially covered with vegetation. The adjacent flood plain is further considered



along this channel section. Main focus within numerical modelling is put on one specific building, situated orografically right in a distance of approximately 17 m to the channel.

Figure 4 presents a perspective view of the considered building and also shows top views of both, the building's basement level and the first floor. With about 130 m² floor area, the building features a rather complex structure, including a couple of

potential openings for flooding, such as doors, windows and light shafts. With regard to the numerical model (Sect. 2.4) and the analysis of the simulation results (Sect. 2.5), the structural elements of the building are labelled accordingly. Further information on the structural elements of the building and the potential openings for indoor flooding processes is given in Sect. 2.3 (Table 1).

## 2.3 Hazard and building scenarios

A 300 yr torrential hazard event is considered within hydrodynamic numerical modelling. In accordance with the reconstruction analysis of the flood event in November 2012 and further hydrological analyses of the catchment (Department of Hydraulic Engineering, Autonomous Province of Bolzano, unpublished), the corresponding peak discharge amounts to 120 m³s$^{-1}$. The simulations are accomplished in an unsteady mode, approaching the expected 300 yr flood hydrograph. Due to the computational effort, the simulations do not cover the entire design hydrograph. The investigation focuses on the

rising limb of the design hydrograph, when the discharge exceeds 30 m³s$^{-1}$, and continue till the discharge falls below 30 m³s$^{-1}$ again in the falling limb. A discharge of 30 m³s$^{-1}$ amounts to roughly 60 % of the HQ5-discharge and already leads to initial flooding of the cycle track at the bridge (Hofer, 2014). In order to keep the computational effort for the unsteady model simulations manageable, the simulation hydrograph is chronologically scaled by a factor of 0.1 compared to the expected flood hydrograph under prototype conditions. With it, the computation time for the unsteady hazard scenario is

1020 s and the total discharge volume entering the computational domain amounts to 720,270 m³.

Representing a preliminary study to this unsteady hazard scenario, steady-state simulations with the discharges 87 m³s$^{-1}$, 104 m³s$^{-1}$ and the 300 yr peak discharge 120 m³s$^{-1}$ were also accomplished. Simulation results and details on that are presented by Hofer (2014) and the following discussion of simulation results (Sect. 3.2) merely gives a very brief summary of it. Further, the influence of the simulation mode or rather the considered hazard scenario on the fluid-building-interaction is analysed by

qualitatively and quantitatively comparing the results of the unsteady and steady-state simulations.

Concerning the implementation of the considered element at risk in the numerical model, three scenarios are analysed. Each is characterised by a certain degree of mutual influence between the building and the flow field on the adjacent flood plain: Scenario (a) treats the building as a "fully blocked structure", not enabling any flooding processes. The building envelope is thereby in accordance with the perspective view in Fig. 4. All doors, windows and light shafts are permanently blocked.

Table 1 illustrates the features of the wall elements e1-e7 on the first floor of the building envelope (Fig. 4). Concerning the listed wall areas, the dimensions of windows and doors are not included therein, although assumed to be closed for this scenario. Reflecting current standard practice and methods in flood risk management and, more specifically, the




consideration of buildings within inundation mapping (e.g. Tsakiris, 2014; Habersack et al., 2007), scenario (a) with a fully blocked element at risk represents to some extent a reference for further scenarios.

With scenario (b), the building is treated as a "permeable structure". Doors, windows and light shafts are permanently and entirely open. This assumption runs contrary to scenario (a) but, however, does also not fully conform to typical natural

conditions. Also for scenario (b), the features of the wall elements (first floor) are listed in Table 1. In this case, wall surfaces inside the building and on the outside are separately considered with components each in the numerical model in order to allow for an individual analysis of wetted areas and fluid forces acting on the walls.

As shown in the perspective view in Fig. 4, the building features a couple of openings on its southwest- and west-side, both directly facing to the Rio Vallarsa torrent channel. Dealing with the efficacy of local structural protection measures, scenario

(c) further considers specific permanent modifications at the building, which are intended to reduce or best possibly prevent the fluid from entering und perfusing critical spots of the building. There, the light shafts s1, s4 and s5 (Fig. 4) are closed with a cover and the top levels of the light shafts s2 and s3 are raised by 0.8 m to a level which is expected to overtower the critical flow depth on the adjacent flood plain. Remaining openings of the building envelope are considered to be open which is in accordance to the setting of scenario (b). Concerning the results of scenario simulation it is intended to point out the

influence of local structural protection measures on the spatial and temporal progression of fluid influx. Basically, a full prevention of fluid influx into the building with the measures tested in scenario (c) is not expected.

For all building scenarios (a), (b) and (c), both the steady-state events and as well the unsteady torrential hazard scenario are computed.

## 2.4 Numerical model

Hydrodynamic numerical modelling is accomplished with the FLOW-3D software (Flow Science Inc., 2012). The model scheme and a perspective view of the FAVOR-model (Flow Science Inc., 2012) are illustrated in Fig. 5.

The computational domain, basically covering the section of the brick work canal of the Rio Vallarsa (Sect. 2.2) and the adjacent flood plain orografically right to the channel, is meshed with six structured, orthogonal mesh blocks (mb). The grid resolution is equally set to 0.167 m x 0.167 m x 0.167 m for every mesh block. The input boundary is defined as a bottom

inlet, represented by two small and accurately defined areas at the upstream model boundary and inflow velocities in a positive vertical direction. At the model outlets on the flood plain, pressure boundary conditions are set, each with the assumption that unrealistic backwater effects can be excluded. As illustrated in the model scheme in Fig. 5, pressure boundary conditions are set at the Xmin-, Ymin- and Ymax-boundary of mesh block mb2. At the downstream edge of the paved channel, the boundary condition "outflow" is applied. Concerning both, the grid resolution and the boundaries,

comprehensive tests on their influence on the flow field within the computational domain has been accomplished by Hofer (2014). With regard to accuracy and computational effort, the mentioned grid resolution offers an optimal compromise and the boundary "outflow" copes best with a varying discharge at the ungaged downstream edge of the channel (Hofer, 2014). Mesh block mb6 is set due to the fact that in case of higher discharges, the flow enters also the cycle path in the near range



of the bridge. Mesh block mb6 allows for a spreading along the cycle path towards upstream without reaching the Ymax-boundary of the mesh block.

The considered element at risk is situated within mesh block mb2. Depending on the considered building scenario (Sect. 2.3), it is modelled as blocked or permeable structure or rather a structure with local structural protection measures. In order

to individually analyse wetted areas and force magnitudes on the wall elements, every element is implemented as an individual component in the software. Further, the distinguish between impacts inside the building and on the outside, the wall elements of the first floor are modelled with two components each, partially overlapping each other and shaping the wall structure together (Hofer, 2014). The remaining buildings and objects are modelled as blocked objects. With it, the 3D-numerical model contains 7.05 millions of cells. Thereof, 2.65 millions of cells represent active cells for the simulation (for

scenario (b)). Surface roughness and vegetation are considered adequately, the chosen additional roughness coefficients are mentioned in Fig. 5. Concerning the turbulence options in the numerical simulations, the standard two-equation k-$\varepsilon$- turbulence model is set.

## 2.5 Results of unsteady hydrodynamic modelling

Figure 6 illustrates snap shots of the simulation for scenario (b) with the assumption of a permeable building envelope.

Perspective views of the computational domain at four different time frames are pictured. The colouring of the fluid isosurface denotes to the total hydraulic head, which includes water depth and velocity head. The isosurface value is thereby set to 0.25 in order to illustrate very low water depths on the outer channel embankment. Further, the flow rates at the channel in- and outflow of mesh block mb1 (Fig. 5) point out the maximum discharge capacity in the channel and the fluid volume impacting the adjacent flood plain. Negative flow rates at the mesh block boundaries are due to the orientation of the

coordinate system set for the computations.

Generally, overtopping of the brick work channel banks appears approximately at 60 m³s$^{-1}$ discharges, flooding occurs mainly at the outside of the channel bend immediately after the bridge crossing. This basically confirms to observations during the flood event in November 2012 (Sect. 2.2). After roughly 285 s of simulation, flooding initially reaches the building envelope and starts wetting. Due to the enclosing wall of the neighbouring building, flooding is to some extent

deflected to the south- and southwest-faces of the building, at least when the flooding process has not made much progress yet. With it, perfusing of the building is initially observed at the light shafts s2 and s3. The basement level is filled and after around 450 s of simulation, flow depths significantly increase also in the first floor of the building. The flow exits the building mainly via the light shafts s4 and s5 from the basement level and the openings of wall element e7 on the first floor. Within the falling limb of the hydrograph, the flow depths in the building and on the adjacent flood plain decrease again.

However, the basement level of the building remains fully filled up to the level of the storey ceiling. It should be noted that the storey ceiling is not figured out in Fig. 6, it is of course considered within numerical modelling. Wall elements of the basement level are coloured red in Fig. 6, those of the first floor have a white colour.





To give a further impression on the characteristics of influx into in the building structure and flow conditions inside, Fig. 7 illustrates depth averaged velocities at sections in the directions of the x- and the y-axis, again for scenario (b). With the section in the direction of the y-axis as a spatial reference, streamlines depict main flow paths at different time frames during simulation. A rather turbulent and temporarily significantly changing flow pattern characterises the situation within the building. Initially, as long as the basement level is not entirely filled, the fluid enters the building mainly via light shaft s3 and the flow field in the building has the distinctive rotational character. With progressing simulation time, the flow pattern becomes more and more disordered and flow in both directions occurs at the openings of the building envelope and inside. Initially, maximum depth averaged velocities up to 5 m/s occur inside the building. These maxima are spatially limited to the vertical drops at the light shafts. Subsequently and, if focusing on the conditions on the basement level, with increasing filling ratio of the building volume, flow velocities significantly decrease and approach almost zero values.

In the following, the simulation results for building scenario (b) are compared with those for scenario (a), the reference case with the blocked building. An analysis is firstly made for the wetting progress at the outside of the building envelope. Figure 8, top line, illustrates the chronological sequence of the wetted area / total area ratios exemplarily for the wall elements e1, e5, e6 and e7 (Fig. 4 and Table 1). Accordingly, the block diagrams in the lower line in Fig. 8 point out the wetting durations. The time-dependent ratios wetted area / total area of the wall elements are thereby classified and the number of simulation output time steps à 10 s are analysed. Results for scenario (a) are coloured in black, the colour red is assigned to the results for scenario (b).

Concerning the peak ratios, there is only a marginal difference between the scenarios (a) and (b). A maximum wetting percentage of roughly 25 % occurs at wall element e1 for both scenarios, the peaks at e5, e6 and e7 are 50 %, 50 % and 7.5 % accordingly. The comparatively low wetted areas at the outside of wall element e7 are due to the fact that it is oriented to the south of the building at thus not directly exposed to the flow.

However, some significant differences between the two scenarios (a) and (b) can be observed in the temporal development of wetting. In case of scenario (a), flow on the almost flat flood plain is prevented from entering the building at the light shafts (s2 und s3). The wetting ratio at wall element e6 features significantly higher values during the rising limb of the flood hydrograph. The plateau in the red line for wall element e6 until the time frame of 450 s marks the filling progress of the basement level. Once completely filled, water accumulates at the outside of wall element e6 and the wetting ratio increases rapidly. There is no significant difference at wall element e6 between the scenarios during the falling limb of the hydrograph, except for a marginal lower water level for scenario (b) at the end of the simulation. The same holds for the characteristics of wetting at wall element e5. In accordance to the situation at wall element e6, e1 is also impacted more significantly during the rising limb of the hydrograph. Due to the blockage of the building, damming on the flood plain appears earlier and the flow depths at wall element e1 increase accordingly.

As aforementioned wall element e7 is not directly exposed to flooding. The fluid impact is higher for scenario (b) when the basement level is entirely filled and the fluid also exits through the openings of wall element e7. With it, the relative difference in wetting between both scenarios is highest at the building envelope not facing the Rio Vallarsa channel. With



regard to the comparison in the block diagrams, durations with lower wetting ratios are on an average higher for scenario (b), whereas higher wetting ratios are lower. This holds for the wall elements e1, e5 and e6. Concerning wall element the situation is vice versa.

On the basis of the hydraulics at the building, the dynamically impacting fluid forces are analysed in Fig. 9, left. Force
magnitudes at the first-floor-wall elements e1-e7 are compared for the scenarios (a) and (b). Concerning scenario (b), the impacts in- and outside the building are plotted individually (red dots in Fig. 9, left). Force magnitudes are calculated from the temporarily varying pressure and shear forces; they represent the maximum total force on the wall element within the entire simulation period.

Firstly focusing on the outside of the building, maximum impacts with values in the range 22-28 kN occur at the wall
elements e3 and e5. The force magnitudes at the remaining wall elements reach 10 kN at maximum. At the wall elements e1, e4, e6 and e7, only a marginal difference between the two considered scenarios can be observed. At wall element e2, the impacting force for scenario (b) accounts for 35 % of the force for scenario (a). On the contrary, it is 20 % and 13 % higher for the wall elements e3 and e5. Impacting forces inside the building are in general lower than on the outside. This is mainly due to the facts, that (i) the fluid firstly fills the basement level and only insignificantly impacts the first floor at the inside at
the beginning of the flooding and (ii) the force components due to the dynamics of the fluid (lower velocities) are comparatively lower.

With regard to scenario (b), Fig. 9 further shows the wetted area in function of the time each for the wall elements e1-i, e5-i, e6-i and e7-i on the first floor and the corresponding wall elements on the basement level. Concerning the latter, wetting ratios reach a value of 1.0 after 450 s of simulation and remain constant until the end of simulation. Due to the characteristics
of the building structure, wetting on the first floor starts after 450 s of simulation, rapidly reaches its maximum after 470-530 s of simulation and certainly decreases until the end of simulation. The present interaction between the fluid bodies on the basement level and the first floor can be observed in Fig. 9, right: Maximum specific forces, meaning the ratio of fluid force magnitude and wetted area, appear at the time of the maximum flow depth on the first floor. The decrease in water level during the falling limb of the hydrograph leads to a decrease of hydrostratic pressure and, consequently, the specific forces.
However, compared to the water body on the basement level, the influence of the fluid in the first floor is relatively small.

If changing the characteristics of doors, windows and light shafts of the considered element at risk, the fluid impact inside the building is significantly different, not necessarily going along with an exclusive decrease of impacts if specific local structural protection measures are built. This aspect is shown in Fig. 10 by means of a comparison of the scenarios (b) and (c). Wetted areas of the wall elements e1-i, e5-i, e6-i and e7-i on the first floor are compared (left diagram). The situation on
the basement level is shown in the middle and left diagram.

In case of scenario (c), the process of flooding and perfusing the building occurs in a way other than for scenario (b): The initial fluid influx via the light shafts is disabled due its covering and raise. The fluid enters the building through the doors and windows on the first floor, stays and spreads in the building and partially leaves again. The basement level is filled from the fluxes inside via the staircases; it does not become fully filled during the entire simulation period. Accordingly, wetted



areas and impacting forces on the basement level are significantly lower for scenario (c) than for scenario (b). Higher impacts occur for scenario (c) on the first floor, except for wall element e5-i. The latter is affected only marginally due to the facts that staircase sc2 is placed directly in front and the fluid on the basement level does not reach the storey ceiling. Within this context it has to be noted that the scenarios (b) and (c) do not fully accurate the pure natural behaviour of the building in

case of flooding. Doors and windows are assumed to be fully open during the entire duration of the flood hydrograph. At real conditions, if not protected with specific sealing and reinforcement features, they are expected to have neither a fully blocking nor an open but a partially transmissive effect. However, simulation scenario (c) highlights the need of an excellent planning procedure for building hazard-proof buildings in order to achieve efficient and reliable flood protection for the considered element at risk.

Whereas from a building's stability and durability point of view the impact of flooding is of basic relevance, the way of considering a certain element at risk within numerical modelling seems to insignificantly influence the flow field on the adjacent flood plain. To give an impression on this process of interaction, Fig. 11 illustrates time-dependent flow data at the boundaries of mesh block mb2. The differences of discharges between scenarios (b) and (a) (red lines in Fig. 11, left and middle) and as well between scenarios (c) and (a) (blue lines in Fig. 11, left and middle) are related to the maximum

boundary outflow for scenario (a) and plotted as absolute values against time at the boundaries Xmin, Xmax and Ymin of mesh block mb2. The latter covers the flood plain orografically right to the channel (Fig. 6) where a certain influence of the building may be expected.

The relative differences between the scenarios (b) and (a) are 2.7 % (Xmin), 0.8 % (Xmax) and 2.2 % (Ymin) on average. Accordingly, the values for scenario (c) compared to scenario (a) are 1.1 %, 1.1 % and 1.9 %. Maxima only sporadically

exceed a value of 5 % at the boundaries Xmin and Xmax, except for the simulation period from 300 s to 500 s at the Xmin boundary. At the Ymin-boundary the maximum relative outflow differences are 13.4 % and 9.1 %. Xmin represents the inflow boundary for mesh block mb2. Thus, Fig. 11 points out that a minor influence of the building on the adjacent flow field not only appears in the downstream direction where the building acts as a small retentional element. The building means also an obstacle and influences the flow on the flood plain towards upstream. A comparison of the total discharge

volumes at the outflow boundaries of mesh block mb2, again in relation to the simulation time, are highlighted in Fig. 11, right. Basically, at the Ymin-boundary least water leaves the computational domain on the flood plain, whereas the most appears at the Xmax-boundary. Related to the total discharge volume of the flood hydrograph (720,270 m³), merely 1.1 % pass the Xmax-boundary for scenario (a). At the Xmin- and Ymin-boundaries, the volumes amount to 0.6 % and 0.4 % accordingly. A comparison of the scenarios with each delivers volume-ratios of 1.0002 (Xmax), 0.9579 (Xmin) and 0.9633

(Ymin) for scenario (a) vs. scenario (b). Scenario (c) vs. scenario (b) with the ratios 0.9903 (Xmax), 0.9862 (Xmin) and 0.9729 (Ymin) also shows in an insignificant difference.

Differences in flow parameters (water depths, velocities) between the considered building scenarios are as well small, except for the area inside the building and very close to the building envelope. They are considerable smaller within the computational domain than at the model boundaries.





## 3 Discussion and conclusions

### 3.1 Fluid-building-interaction – general relevance of indoor flooding processes under clear water conditions

The results of hydrodynamic numerical modelling (Sect. 2.5) on the one hand show a rather marginal influence of the building on the flow field on the flood plain and in the channel of the Rio Vallarsa torrent. Due to the small interior volume

of the building compared to the volume of the simulated flood hydrograph, this behaviour is comparatively independent from the way of considering the element at risk within the simulation model (Fig. 11). If not scaling the expected hydrograph (by the factor 0.1 in order to cope with the computational effort) and perfectly simulating real conditions, this influence would even be considerably smaller.

Otherwise, focusing on the impact of the fluid on the building at the inside, a certain impact can be observed. This impact at

the inside is mainly characterised by relatively small flow velocities (Fig. 7) but relatively long wetting durations that basically extend beyond the duration of the hazard event. The impact (Fig. 9) does not threaten the stability of the building (limit states ULS according EN 1990, Sect. 1) but affects the building physics and, with it, the usability (limit states SLS according EN 1990, Sect. 1). The latter may cover also potential serious damage of electrical and in-house installations, furnishing and equipment. This kind of damage will be considerably higher under real conditions, when fine sediments

(suspended load) that pass the debris retention dam, contribute also and get deposited inside the building. A significant impact on the stability of the building probably requires the contribution and consideration of conditions with intense sediment loads (WST and DBF according Heiser et al. (2015) or rather the modelling of any geo-mechanical processes (Sect. 1).

However, with regard to the danger to the life and limb inside elements at risk, numerical modelling under clear water

conditions is highly valuable. The characteristics of flooding the building provide information for evacuation planning or rather non-affected areas during hazard events.

### 3.2 Fluid-building-interaction with different hydrological modelling scenarios – comparison of an unsteady and a steady-state modelling approach

The simulation results presented in Sect. 2.5 exclusively focus on the unsteady hydrological scenario, specified as 300 yr

torrential hazard and design event for flood risk management (Sect. 2.4). The present computational domain and element at risk was already studied by Hofer (2014) in terms of a steady-state analysis of specific hazard scenarios. Hofer (2014) simulated three specific discharges (87 m³s⁻¹, 104 m³s⁻¹, 120 m³s⁻¹) and analysed wetted areas and impacts on the considered building at the end of simulation each when steady-state conditions were achieved. By analogy with the unsteady modelling approach, the three building scenarios (a), (b) and (c) were analysed.

When qualitatively comparing the results of the two different modelling approaches, it is obvious that the process of filling the interior volume of the building cannot be adequately simulated with a steady-state modelling approach. To mention one aspect in this context, the basement level of the building is getting fully filled till the end of simulation independently from





flood discharge, only the required simulation time changes accordingly. However, as the evaluation of results is accomplished solely at the end of the steady-state simulations when the maximum fluid impacts are supposed to appear, this presumed "unnatural" process of filling the building to some extents the distorts the modelling results. On the contrary, with the unsteady simulations, it can be shown that maximum potential impacts may appear at conditions, when both the

basement level and the first floor of the building are filled and the water level on the basement level does not reach the storey ceiling. The time of maximum impacts is thus mainly depending on the characteristics of the flood hydrograph. Further, with regard to an analysis of expected wetting durations, a steady-state modelling approach does not allow any conclusions. Against this background, differences in fluid-impacts resulting from unsteady and steady-state modelling can mainly be observed for scenario (c), where the fluid enters the building but does not entirely fill the basement level. Consideration of

characteristics and volume of the flood hydrograph seems at least equally important as the flood peak. Further, the differences of the results for the three considered building scenarios are less pronounced with the steady-state simulations. A qualitative comparison of the modelling results (with the 120 m³s-1 discharge for steady-state modelling) generally delivers rather variable differences in maximum wetting and fluid forces. For instance focusing on scenario (a) and at wall element e1, Hofer (2014) analysed a maximum wetting percentage of 23 %, which fits well with the wetting peak at this wall

element illustrated in Fig. 8. Further, looking at the wall elements e5, e6 and e7, larger differences arise with maximum wetting ratios of 52 % (unsteady) vs. 33 % (steady-state), 48 % (unsteady) vs. 14 % (steady-state) and 3 % (unsteady) vs. 1 % (steady-state). Focusing on scenario (b), the comparison of maximum wetting ratios delivers 24 % (unsteady) vs. 21% (steady-state) for wall element e1-o, 50 % (unsteady) vs. 31 % (steady-state) for e5-o, 46 % (unsteady) vs. 14 % (steady-state) for e6-o and 7 % (unsteady) vs. 3 % (steady-state) for wall element e7-o. An underestimate of the expected peak

values seems evident here for the steady-state simulation. Further details and results of the steady-state simulations can be found in Hofer (2014).

Regardless from capabilities and constraints of both modelling approaches it has to be noted that the steady-state simulations require a substantially lower computational effort. This is due to the shorter simulation times till the simulations reach steady-state conditions and the generally higher numerical effort for the unsteady simulations.

## 25 3.3 Computational modelling effort – capabilities and limits for practical application in flood risk management

Under given effort of hydrodynamic numerical modelling and with consideration of the illustrated intensity of the fluid-building-interaction (Sect. 2.5, 3.1 and 3.2), the basic question, whether there is any sense in considering building flooding processes in practical application, arises. Figure 12 provides performance details from the unsteady numerical calculations. The left diagram shows the computation times for the scenarios (a), (b) and (c), each in relation to the simulation time.

Accordingly, the middle diagram delivers the applied time step sizes and the diagram on the right points out the computation times per time step in relation to the simulation time.

Basically, all accomplished computations require long computing times compared to the simulation time or rather real time conditions. With the use of an Intel core i7-3820 quad-core processor (@ 3.60 GHz), 32 GB main memory and a parallel



software license code, computation times of about 200 hrs are achieved for the scenarios (a) and (b), without any substantial differences between these scenarios. The time step sizes generally decrease with the occurrence of flooding and spreading on the flood plain. The computation time per time step features an almost linear relation to the fluid surface area within the computational domain, again with an insignificant influence of the building flooding process.

With 425 hrs computation time, simulation of scenario (c) is much more costly. Reason therefore is the flow characteristics inside the building: The time step size significantly decreases when the fluid enters the building after about 320 s of simulation time. In contrast to scenario (b), the filling of the basement level occurs via the staircases from this point in time. The flow on these fine-structured, stepped obstacles leads to an adjustment of the time step size.

In a more general sense, 3D-hydrodynamic modelling of flood hydrographs and the spreading on a flood plain is a very time-
consuming task, even though rather small computational domains (0.564 ha in this study) are analysed. The computational effort can but does not necessarily must further increase if considering building flooding processes. This statement is underpinned by the fact, that for the unsteady scenario simulation the expected design flood was adapted by a scale factor of 0.1 (Sect. 2.3) in order to achieve manageable computation times anyway. Practical application in flood risk mapping, typically covering a larger extent of the flood plain and at least a couple of elements at risk, seems to be not practicable (and
mandatory) in this context. Compared to the rather small influence of an open building structure on the flow field on the flood plain, a potential significant increase of computation time is furthermore not reasonable.

However, for vulnerability analysis and the planning of local structural protection measures for certain elements at risk, the modelling of building flooding processes means a valuable tool. Specific planning options can be tested and verified on their efficiency; they can be further compared with each other within a cost-benefit-analysis where potential hazard impacts or
rather avoided impacts and thus damages are considered. Compared to the general expense of the planning process and, for the case of an insufficient efficiency of the measures due to a poor planning, to the extent damages, the costs for numerical modelling and scenario simulation is perfectly acceptable.

## 4 Aspects of further research

The assessment of the specific vulnerabilities of the built environment is the pillar for any planning process that is targeted at
a reduction of the expected adverse consequences. These adverse consequences result from the interactions between the hazard processes and the exposed elements, both in time and space. From a physical perspective, these interactions firstly take the form of damage generating mechanisms, which are quantifiable knowing the hazard intensities and the physical response of the structures in terms of (i) deformations with respect to the admissible states, (ii) the wetting process of the buildings envelope and its alterations. Given specified loading conditions and determined geometrical and material
properties of the buildings envelope, the subsequent mass transport processes through it may result in secondary damage generating mechanisms.



This work represents a step towards the development of a comprehensive physical vulnerability assessment framework and shows for pure water floods only how advanced modelling techniques may be usefully employed. However, further research efforts are needed (i) to develop reliable and practicable 3D codes for the whole spectrum of flow processes involving sediment transport at various rates and featuring different non-Newtonian flow behaviours, (ii) to couple the simulation of

flow dynamics with structural mechanics. In parallel, if the aforementioned advances are feasible, it is fundamental to provide for methods to generate building models close range photogrammetry. Additionally, it is essential to optimize the physical parameterization (i.e. material properties) of such models. In this context, physical scale model experiments could provide novel and valuable insights.

Self-evidently, only a limited number of key elements of the built environment should be analysed to such a level of detail,

therefore a harmonization with the available vulnerability information for the remaining exposed elements is necessary.

To conclude, it has to be considered that vulnerability is conceived as a continuum along the risk cycle with respect to both, time and space, and that derived knowledge can be explored to identify priorities for action towards an increased societal resilience. The derived societal knowledge, however, always is a convolute of objective and subjective elements. In fact, the set of the former is incomplete and may exhibit significant levels of uncertainty, due to the inherent randomness of natural

phenomena. The latter, though valuable, is often expressed through underspecified and qualitative argumentation strings. Therefore, specific research should be devoted to promote knowledge convergence between the different disciplinary domains.

### Acknowledgements

The presented study was accomplished at the University of Innsbruck and in close collaboration with the Department of Hydraulic Engineering of the Autonomous Province of Bolzano. The Unit of Hydraulic Engineering thanks the Department of Hydraulic Engineering for this precious cooperation. Further thanks are due to the owner of the modelled element at risk for providing highly valuable plan documents of the building structure and experience of flood event characteristics, and, most notably, keeping an open mind for scientific research.

The presented study was conducted during complaint stage of a three-years research project, funded by the Austrian Science Fund (FWF): P 27400-NBL. For it, it provides valuable information on capabilities and limitations of numerically modelling the process-building-interaction.

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





Table 1. Building representation for the considered scenarios (a), with a fully blocked structure, and (b), assuming all doors, windows and light shafts entirely open; wall element notations refer to Fig. 4; for scenario (b), index "o" means the outside of the wall element, "i" refers to the inside.

Building representation – first floor

| | Wall element | Total area [m²] | Doors Number | Doors [m²] | Windows Number | Windows [m²] | Wall area [m²] | Wall length [m] |
|---|---|---|---|---|---|---|---|---|
| **Scenario (a), blocked structure** | e1 | 13.44 | 2 | 3.84 | 1 | 1.09 | 8.51 | 2.20 |
| | e2 | 7.46 | 1 | 2.02 | - | - | 5.44 | 1.33 |
| | e3 | 19.05 | - | - | 2 | 2.34 | 16.71 | 5.95 |
| | e4 | 1.91 | - | - | - | - | 1.91 | 0.59 |
| | e5 | 13.44 | - | - | 1 | 1.19 | 12.25 | 4.20 |
| | e6 | 33.28 | - | - | 2 | 3.98 | 29.30 | 10.4 |
| | e7 | 46.03 | 4 | 13.30 | - | - | 32.73 | 7.73 |
| **Scenario (b), permeable structure** | e1-o | 13.44 | 2 | 3.84 | 1 | 1.09 | 8.51 | 2.2 |
| | e1-i | 13.73 | 2 | 3.84 | 1 | 1.09 | 8.80 | 2.36 |
| | e2-o | 7.46 | 1 | 2.02 | - | - | 5.44 | 1.33 |
| | e2-i | 6.02 | 1 | 2.02 | - | - | 4.00 | 0.88 |
| | e3-o | 19.05 | - | - | 2 | 2.34 | 16.71 | 5.95 |
| | e3-i | 16.48 | - | - | 2 | 2.34 | 14.14 | 5.15 |
| | e4-o | 1.91 | - | - | - | - | 1.91 | 0.59 |
| | e4-i | 2.05 | - | - | - | - | 2.05 | 0.64 |
| | e5-o | 13.44 | 1 | - | 1 | 1.19 | 12.25 | 4.20 |
| | e5-i | 12.80 | 1 | - | 1 | 1.19 | 11.61 | 4.00 |
| | e6-o | 33.28 | - | - | 2 | 3.98 | 29.30 | 10.40 |
| | e6-i | 30.72 | - | - | 2 | 3.98 | 26.74 | 9.60 |
| | e7-o | 46.03 | 4 | 13.30 | - | - | 32.73 | 7.73 |
| | e7-i | 42.50 | 4 | 13.30 | - | - | 29.20 | 6.63 |




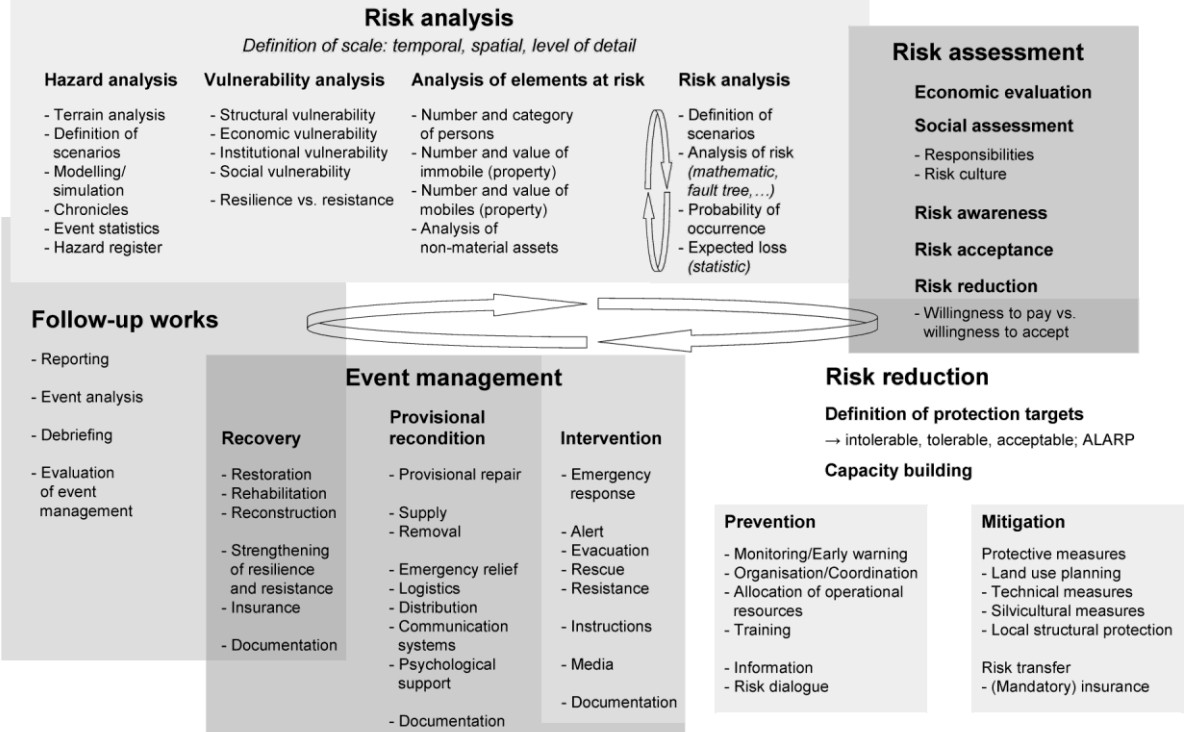

Figure 1. Model of integral risk management conceptualised as "risk cycle" (adapted from Kienholz et al. (2004); Alexander (2000); Carter (1991).



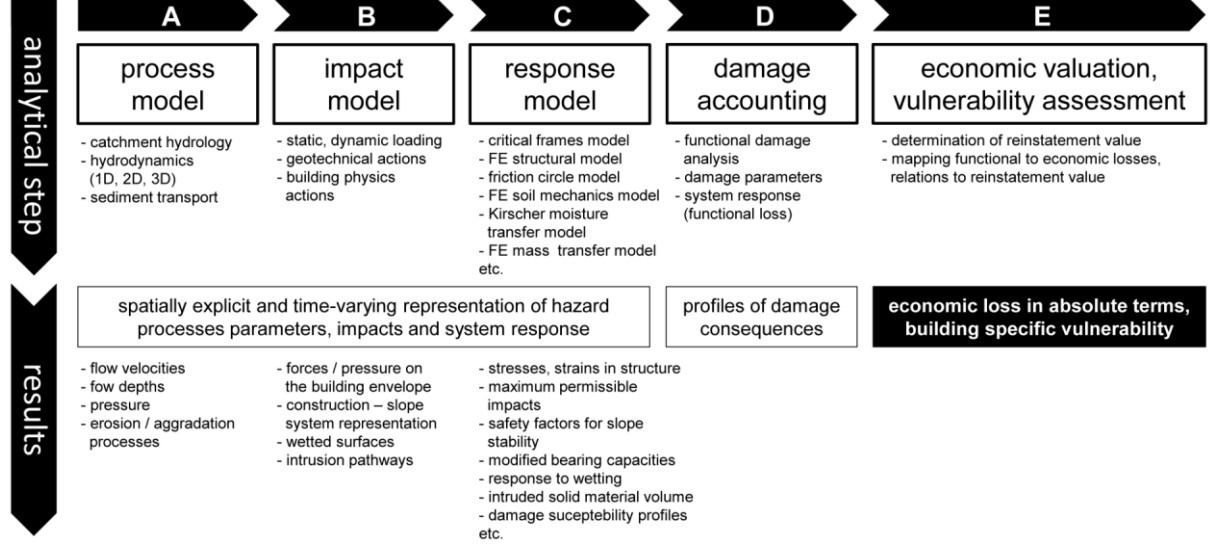

Figure 2. Proposed physics-based vulnerability assessment scheme according Mazzorana et al. (2014) (modified).




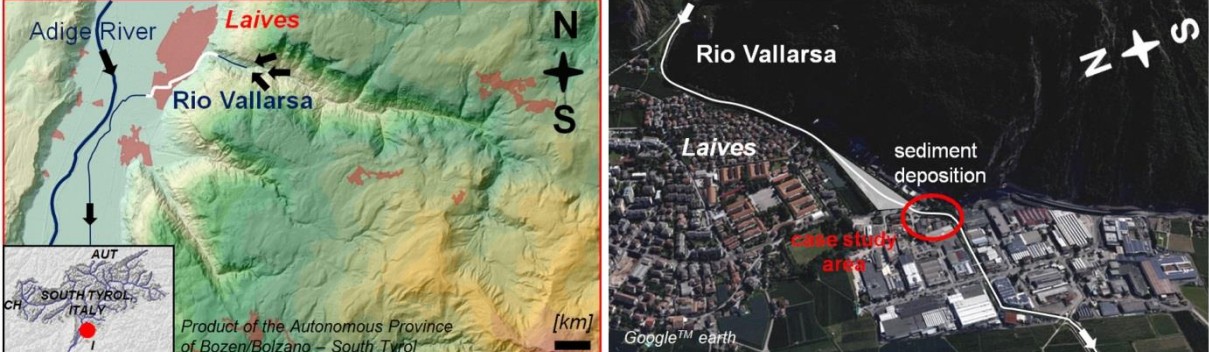

Figure 3. (left) Overview of the Rio Vallarsa catchment and the Adige valley at the village of Laives in South Tyrol, Italy; the colour scheme characterises an elevation model with a 2.5 m-interval; (right) track of the Rio Vallarsa torrent channel

5   through settlement and commercial area in the south-west part of Laives; location of the case study area on the alluvial fan.





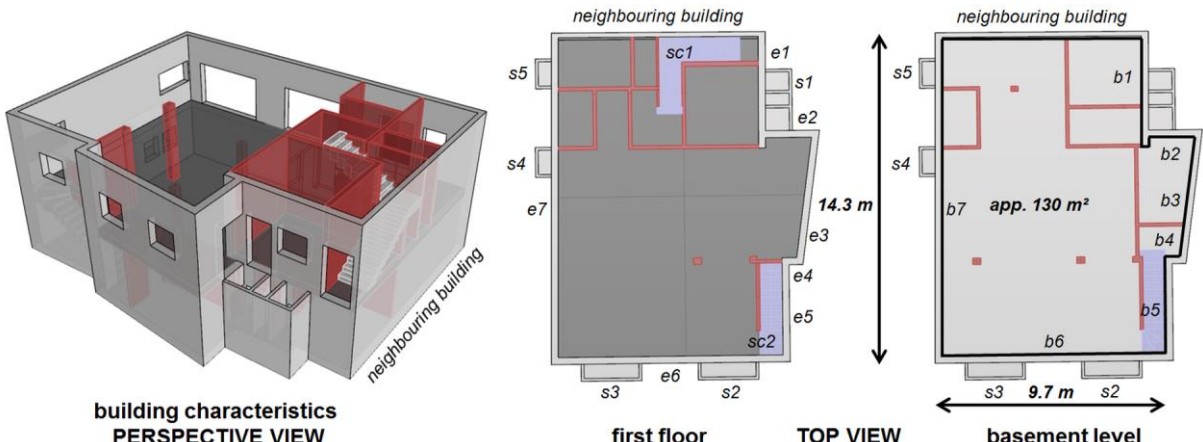

Figure 4. Object of investigation within 3D-hydrodynamic modelling – (left) perspective view and (middle, right) top views of first floor and basement level; notations e1-e7 identify exterior walls on the first floor, b1-b7 denote to corresponding wall elements on the basement level; inner walls are coloured red, s1-s5 characterise light shafts and sc1-sc2 identify stair cases (coloured blue).




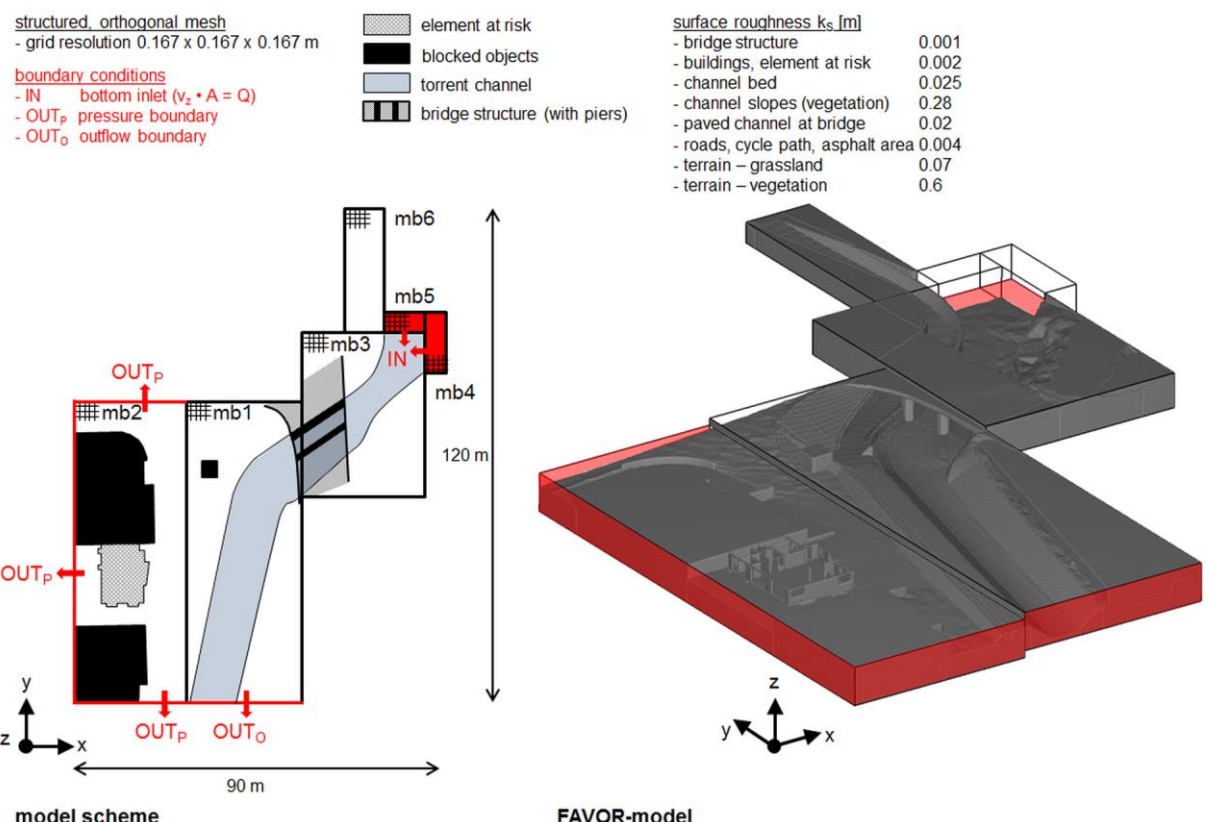

Figure 5. (left) Scheme of the hydrodynamic numerical model; (right) computational model representation with the FAVOR-method (Flow Science Inc., 2012).




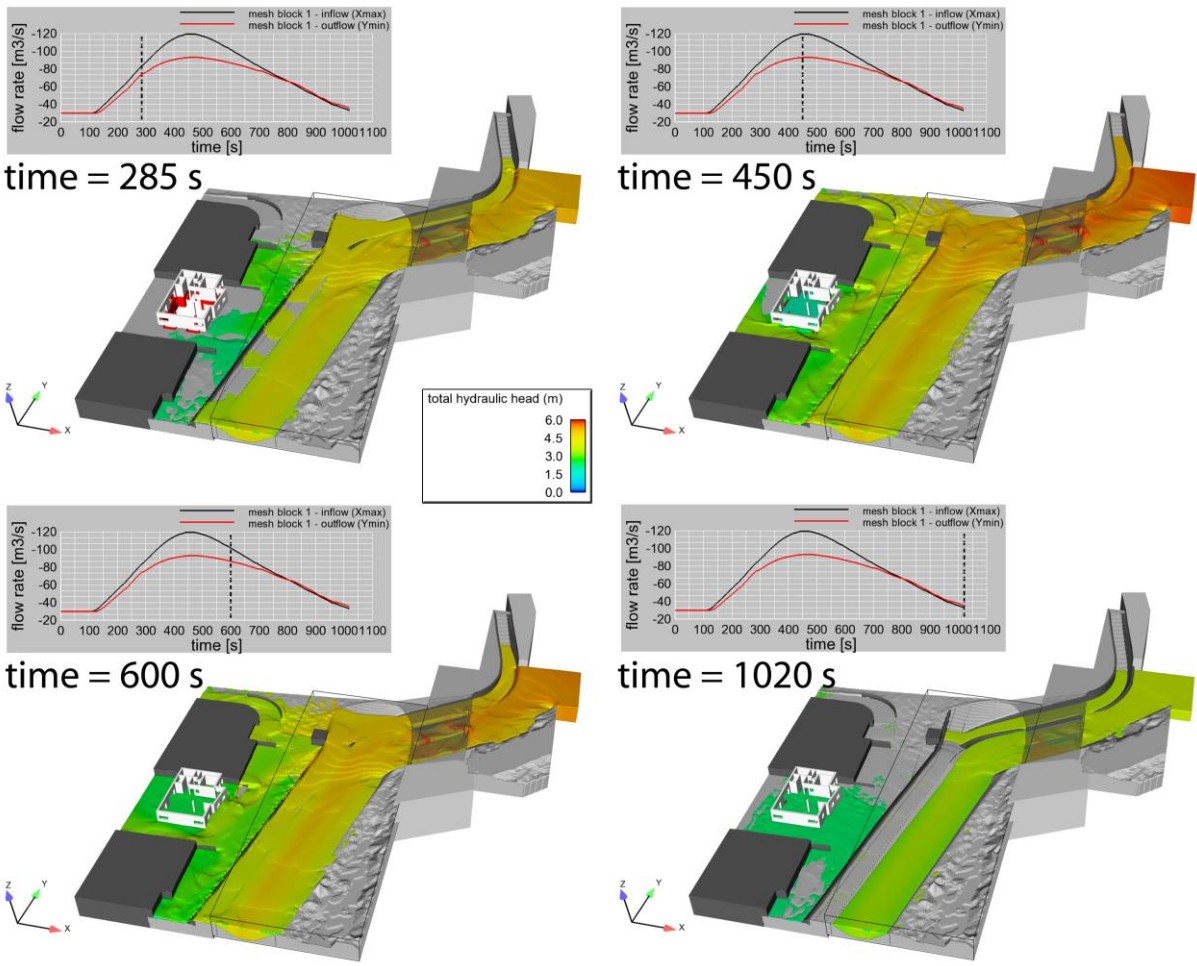

Figure 6. Modelling results for scenario (b) after 285 s, 450 s, 600 s and 1020 s of simulation – perspective view of the stl-geometries and the fluid isosurface representing the total hydraulic head [m], boundary flow rates (Xmax, Ymin) for mesh block mb1.





Figure 7. Scenario (b) after 350 s, 450 s, 600 s and 1020 s of simulation – 2D-sections and streamlines with depth averaged velocity contours [m/s], illustrating flow paths and characteristics of in- and outflow of the building (stl-geometry of the building).





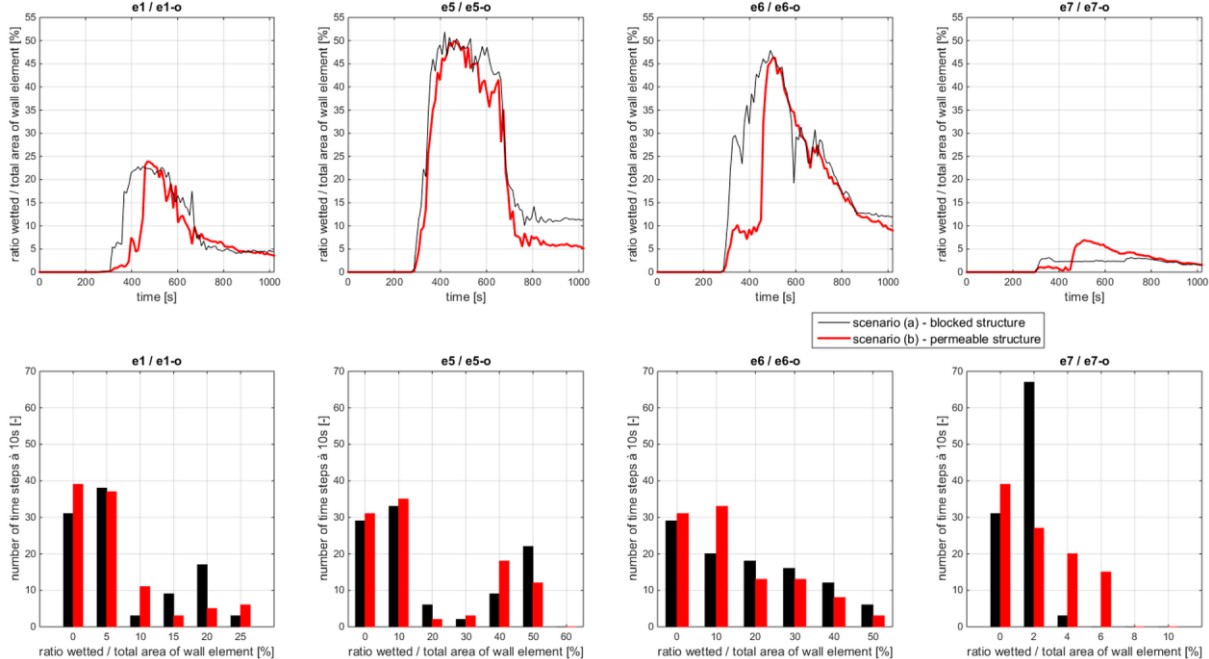

Figure 8. (top line) Ratio of wetted and total area of the wall elements e1, e5, e6 and e7 at the outside of the building for the scenarios (a) and (b); (lower line) comparison of wetting durations (number of time steps à 10 s) for the considered wall elements.





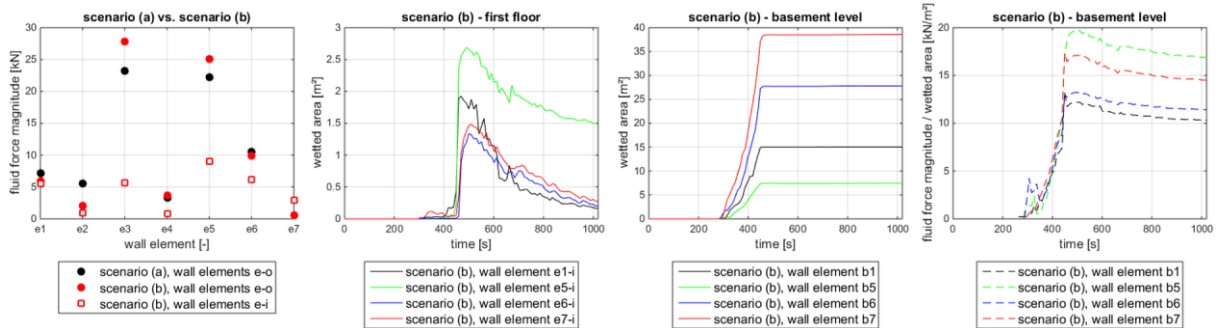

Figure 9. (left) Fluid force magnitudes [kN] on the building envelope for the scenarios (a) and (b); (middle) wetted areas of wall elements [m²] inside the building for scenario (b); (right) specific fluid force magnitudes [kN/m²] at wall elements on the basement level for scenario (b).





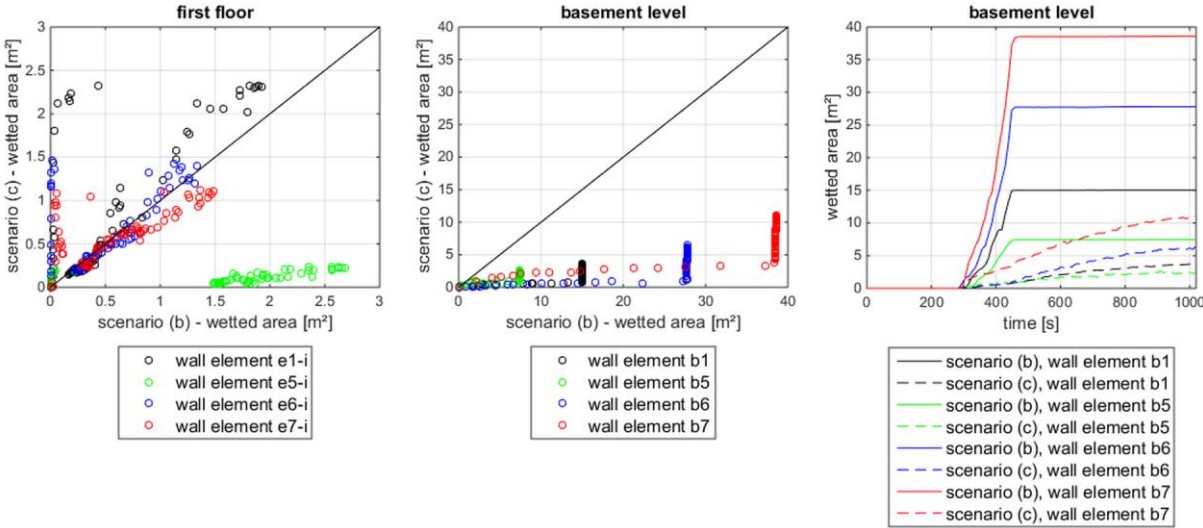

Figure 10. Comparison of wetted areas [m²] during the simulations of the scenarios (b) and (c) – (left) wall elements e1, e5, e6, e7 on the first floor inside the building; (middle) wall elements b1, b5, b6 and b7 on the basement level (simulation results with output time steps à 10 s); (right) temporal characteristics of wetting for the wall elements on the basement level for the scenarios (b) and (c).





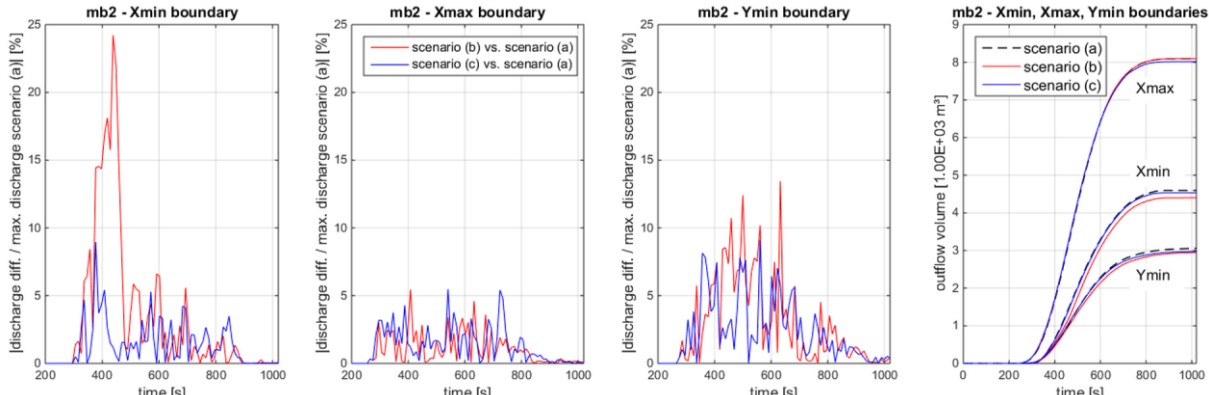

Figure 11. Discharges at the boundaries Xmin, Xmax and Ymin of mesh block mb2 (Fig. 5) – (left and middle) time-dependent discharge differences between scenario (b) and scenario (a) in relation to the maximum discharge of scenario (a) at the considered boundaries; (right) scenario-specific outflow volumes at the boundaries.





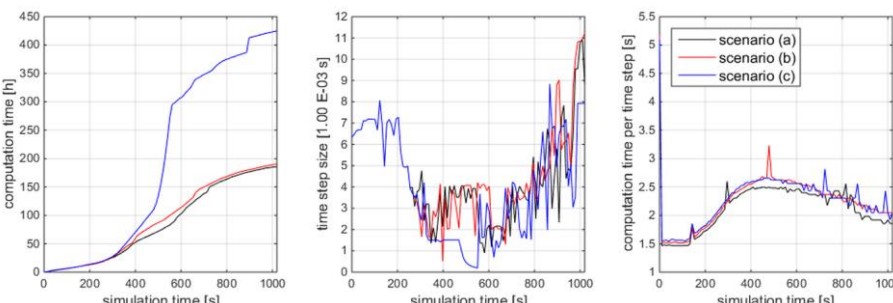

Figure 12. Computational effort – comparison of computation time [h] (left), time step size [1.0 E-3 s] (middle) and computation time per time step [s] (right) for the scenarios (a), (b) and (c).

