# Peer review of "3D-hydrodynamic modelling of flood impacts on a building and indoor flooding processes"

_Natural Hazards and Earth System Sciences, 2015_

## Author Comment (AC1) · 19 Jan 2016

Unfortunately, the caption of Fig. 1 is not fully represented in the discussion paper. Fig. 1 should be read as: "The model of integral risk management conceptualised as "risk cycle" (adopted from Carter 1991; Alexander 2000; Kienholz et al. 2004), based on an earlier version in www.nahris.ch and adopted from Fuchs (2009). "

Reference: Fuchs, S.: Mountain hazards, vulnerability, and risk – a contribution to applied research on human-environment interaction. Habilitation thesis at the University of Innsbruck, Austria, unpublished, 2009.

---

## Referee Comment (RC1) · Anonymous Referee #1 · 19 Feb 2016

I consistently reviewed the paper entitled 3D-hydrodynamic modelling of flood impacts on a building and indoor flooding processes" by Gems, Mazzorana, Hofer, Sturm, Gabl and Aufleger. The manuscript Ref. No. is nhess-2015-326. The paper deals with the assessment of flood flow impact on a single building with respect to wall and floor openings. The analysis is based on 3D numerical modelling using flow3D. Different scenarios have been simulated, among stationary as well as transient flood discharge. Only clear water is considered. The building is located on the flood plain near the torrent. An important point is that there is a bedload retention basin upstream of the site, therefore, bed load and coarse sediment transport can objectively be neglected in the simulation. The paper is well structured and the reader can follow it from A to Z without difficulties. The presented study addresses relevant scientific and technical questions within the scope of NHESS. The only numerical approach may be criticized, but no

on-site measurement or past damages have been assessed. There is not a strong theoretical base on this type of loading on a building (but one might look into the national codes on these types of loads on buildings, maybe as well from Tsunami resistance), the chosen approach be considered judicious. The paper present a more or less novel concept, but it is slightly away from actual research, and potentially a high-quality consultancy firm could do the work as well. The in depth analysis and presentation of the results can be considered as relevant for the journal. The numerical model set-up is sufficiently explained and presented; the used references are consistent with the subject. The introduction chapter dealing with the vulnerability assessment within integral flood risk management presents the state of the art. The simulation and main results are sufficient to endorsement the analyses and the conclusion. The description of the data used, the numerical methods and main assumptions and the results obtained are adequately and comprehensive; peers should be able reproducing the main procedures. The most relevant aspects are well explained and mastered with concise literature references. There are no particular questions that need to be considered in addition. The use of the English language is of sound quality. The proposed title allows a good start point to the manuscript, and the reader should not be disappointed. Figures are adequate and sufficiently explained. A more in depth discussion on the results and conclusions to be drawn for new buildings in the same situation is lacking, as well as indication on evacuation of persons or early warning. The paper never mentions the uplift on a building that may structurally damage it. I think that this article gives nice work through careful and thoughtful numerical investigations, proposing comprehensive new insight on flow impact on a building in the flood plain. It can be published with minor revision on the points raised in the above paragraph. It has to be mentioned here that no real English proofreading has been performed by the reviewer.
* * *

---

## Referee Comment (RC2) · Anonymous Referee #2 · 4 Apr 2016

This review addresses the paper "3D-hydrodynamic modelling of flood impacts on a building and indoor flooding processes" by Gems, Mazzorana, Hofer, Sturm, Gabl and Aufleger. The manuscript Ref. No. is nhess-2015-326. The paper demonstrates the application of a 3D model on a object at risk and the influence of different scenarios onto the internal flow processes within the building. While the methodology is able to demonstrate the influence of measures the protection scenarios are very abstract. Also the other border conditions are very specific, so it will be difficult to transfer the results to other regions or the effect of measures applied. In some parts the paper is difficult to read (f.e. page 11 and 12) as it refers to the all the numbers of wall segments, not being a reviewer I would skip these sections. It is not clear why the steady state modelling had been performed as it seems clear that the limitation of volume is an important factor in the filling process of the building. Even damage assessment and cost benefit

analysis are addressed it remains unclear, what type of benefit is expected from the 3D approach compared to a 2D in this domain. As there is even a lack of data for applying damage functions based on 2D data there is nearly no damage information that can be used to really use the additional value of 3D data. Even the simulation results are impressing and seem to be plausible, there would be a need for validation. It can be assumed, that the different materials in a building and the holes for pipes and service lines influence very much the flow into and within the building, this should at least be addressed in a proper way. Also it is not well addressed, why this specific building was chosen, is it a typical one or just the data have been available. How was the grid size chosen in the 3D model? There should be an English prove reading, as there are some mistakes and uncommon use of wording (like reoccurance interval instead of return interval) In general the structure of the paper is fine, after some revisions the paper may be published. Still a vision of using the methodology in practice is not existing or at least the potential not well explained. The conclusions are not convincing to make use of this methodology in near future.

---

## Author Comment (AC2) · 7 Apr 2016

The present authors' comment, referring to the discussion paper titled "3D-hydrodynamic modelling of flood impacts on a building and indoor flooding processes", is aimed at the comment of anonymous referee #2, published on 4 April 2016.

The authors of the manuscript would like to thank the reviewer for the valuable comment, which will lead to an improvement of the manuscript during further revision process. A couple of issues are briefly mentioned, they are commented by the authors as follows:

(1) The tested local structural protection measures are abstract

Building scenario (c) covers a set of structural protection measures which are intended

to prevent the fluid from entering the building, at least to a certain point in time during the considered flood event. Though considering the typical design of object protection measures (see therefore also Hofer (2014)), the set of structural measures was basically chosen arbitrarily. Since the primary focus of our investigation was to evaluate the possibilities and limits of the chosen modelling approach for flood prone buildings on a typical floodplain, working with fictitious scenarios, is, from the author's point of view, a reasonable procedure. The robustness of the modelling approach allows for an ex ante effectiveness test of envisaged structural mitigation options. To summarize, our basic aim was to show the full applicability of the modelling approach also with small scale and local structural changes within the built environment.

(2) The computational modelling results described in the manuscript pages 11 and 12 are difficult to read

The authors agree with this reviewer comment. The computational modelling results on pages 11 and 12 may be difficult to read due to the complexity of the building structure. We chose, for completeness to consider every single wall element. Moreover, the general arrangement of the figures at the end of the manuscript (the text on the pages 11 and 12 refers to the figures 9, 10 and 11) does not support the reader in keeping track of the entire result spectrum.

We will, within the revision process, slightly shorten this description of the modelling results and eventually provide a table which summarizes most relevant numbers mentioned in the text with better clarity and conciseness.

(3) The benefit of steady-state modelling is at least questionable

The authors fully agree with the referee in this context – steady-state modelling means an essential simplification compared to typically unsteady conditions during a flood event in a torrent catchment. Steady-state modelling scenarios have to be carefully interpreted within flood risk management in general, but they are still sporadically applied in practice. With respect to the conditions in the case study area and at the Rio

Vallarsa this is even more true due to the following aspect: The design flood event is characterized by a comparatively short duration and a rather small discharge volume. A rather long duration of steady state modelling would lead to a significant overestimation of the fluid volume entering the building. This aspect is discussed in section 3.2 of the manuscript. However, if focusing on the efficiency of the local structural protection measures and thereby the location and time of initial flooding of the building, a steady-state scenario simulation also provides valuable information for the planner (Hofer, 2014). Due to this fact, the authors decided to provide a short section within the discussion of the modelling results to point out the relevance of unsteady modelling approach.

By again referring to the work of Hofer (2014), the informative value of the steady-state simulations will be more clearly pointed out during further revision process.

(4) Need for an adequate model calibration (validation)

and

(5) The aspect of transferring results of the case study analysis to other regions and / or buildings is not sufficiently discussed

This issues are also mentioned by anonymous referee #1 and the authors comment on that accordingly: To our knowledge previously occurred past events (at the Rio Vallarsa) did not cause relevant damages in the case study area and on the building selected for computational modelling purposes. As stated in section 2.2 of the manuscript, a flood event occurred in November 2012 and it is assumed from the Department of Hydraulic Engineering (Autonomous Province of Bolzano) that the designed stream channel can cope with discharges in the range 30-40 $m^3$/s before overbank flooding occurs. The data and information from the 2012-event led to an adaptation of the HQ100- and HQ300-discharge design hydrographs and this was also considered in the present work (section 2.2), which is primarily meant to analyse an extreme design flood (HQ300) (section 2.3, first paragraph). With the available information, the

computational model was best possibly calibrated by adjusting the surface roughness parameters in the stream channel. As stated in section 2.3 (first paragraph) of the manuscript and already discussed in the work of Hofer (2014), discharges higher than 30 m$^3$/s exceed the channel capacity in the model which fits well with the available information and expert assessment. In summary, the roughness coefficients are well calibrated in the stream channel and – since any observation data is not available – we chose roughness coefficients values for the floodplain and the building structure according to values commonly cited in literature.

Since the aspect of model calibration is also mentioned in the comment of anonymous referee #1, the sections 2.2 and 2.3 of the manuscript are more detailed and clearly formulated within the further revision process.

The authors fully agree with the reviewer's note that consequences and benefits for planning new buildings, potentially resulting from the case study analysis at the Rio Vallarsa torrent, and also potential transfer of results to other regions / objects are not clearly stated in the conclusions. With regard to the computed impacts and wetting durations on the considered building, it seems not reasonable to transfer computed specific impact loads under design flood conditions to any further objects. The modelling results showed that the computed impacts and the flooding inside the building are significantly influenced by the design flood characteristics (hydrology and, if relevant, sediment transport processes), the capacity of the torrent channel and also by the topography of the adjacent floodplain. The general knowledge of a reasonable application of three-dimensional models for simulation of indoor flooding processes and, further, its computational limits represent the actual added value and novelty of the present case study analysis.

Within the further revision process this aspect is pointed out in more detail.

(6) No damage assessment and within this context the question concerning the benefit of a 3D- compared to a 2D- numerical modelling approach

In the context of the proposed physics-based modelling approach proposed by Mazzorana et al. (2014) the presented work firstly focuses very much in detail on the hydraulic modelling aspect and, thereby, on the possibilities of simulating indoor flooding processes. Damage modelling, as it was done by Mazzorana et al. (2014) in a case study with intense bed-load transport and a 2D-numerical modelling approach, was chosen not to be accomplished in a first step. Compared to this case study and with regard to the modelling results under clear-water conditions at the Rio Vallarsa, it could be probably assumed that the computed impacting forces at the Rio Vallarsa will not endanger the stability of the building. They affect the usability of the building, damage inventory and endanger human life. However, damage assessment represents an interesting issue and could be an issue of further research (see therefore also section 4 of the manuscript).

The benefit of a 3D-numerical modelling approach is quite simply to adequately simulate flooding processes around and inside a building. Conventional 2D-numerical models are based on the Saint-Venant-equations and they provide depth-averaged flow parameters. Complex structures, as the building considered in our work, cannot be adequately considered in a 2D-numerical model. They are conventionally considered by a non-permeable building envelope in 2D-numerical models.

(7) Influence of different materials and structures in the building on modelling results

As the terrain and as well the building structure represent rigid elements in the numerical simulation (immobile obstacles within the flow field), any material parameters concerning the statics and stability of the building have no importance in the model. There is one parameter, surface roughness ks, that characterises the surface properties of the obstacles and thus influence hydrodynamic modelling. Figure 5 in the manuscript contains a list of the applied roughness parameters, differentiated according to the surface properties of the obstacles in the computational domain. As also mentioned in issue (4), the parameters are the results of model calibration and / or represent values commonly cited in literature. Different structures in the building and,
more general, different approaches of hydraulic permeability of the building do have an influence on hydrodynamic modelling on the adjacent flood plain and also on the forces and wetting durations impacting the building. To analyse this is one of the main issues of the present manuscript. From the author's point of view, this was substantially analysed by simulating and comparing three building scenarios with different hydraulic permeability.

(8) Choice of case study area and building and the definition of the mesh grid size of the numerical simulation

Executing hydrodynamic modelling (no bed-load transport) as a first step towards the proposed physics-based modelling approach proposed by Mazzorana et al. (2014), the case study area the floodplain at the Rio Vallarsa in Laives (South Tyrol) seems well-suited for further analyses mainly due to the following reasons (they are comprehensively discussed in the sections 2.1 and 2.2 of the manuscript): - General risk of flooding of the case study area in the current state situation during flood events with peaks > 30 $m^3$/s (appr. HQ10) in the Rio Vallarsa; the considered building is thereby directly exposed to the hazard process and does not feature any structural protection measures against indoor flooding processes. - Non-relevance of bed-load transport due to the existence of a retention basin at the upstream boundary of the case study area. - Rather small spatial extent of the rigid torrent channel on the local floodplain in order to perform complex numerical simulation with still manageable computation times. - Availability of data for numerical modelling (topography, building characteristics, hydrology, etc.) and observation data.

Concerning the mesh grid resolution of the numerical model, comprehensive tests have been accomplished by Hofer (2014). This is stated in section 2.4 of the manuscript. The chosen grid resolution (0.167 m x 0.167 m x 0.167 m) leads to sufficiently adequate modelling results and exhibits still challenging computation times (see therefore section 3.3). A further refinement of the computational mesh does not noticeably change the modelling results.

(9) Manuscript language and style

It is found that an English prove reading is necessary before final publication. This issue is also mentioned by anonymous referee #1. Accordingly, the manuscript is again carefully checked within the further revision process.

References:

Hofer, T.: 3D-numerische Modellierung der Durch- und Umströmung von Infrastruktur-objekten (Gebäuden). Master thesis, Unit of Hydraulic Engineering, University of Innsbruck, 2014 (in German). Mazzorana, B., Simoni, S., Scherer, C., Gems, B., Fuchs, S., and Keiler, M.: A physical approach on flood risk vulnerability of buildings. Hydrol. Earth Syst. Sci. 18, 3817-3836, 2014. Doi: 10.5194/hess-18-3817-2014

---

## Author Comment (AC3) · 7 Apr 2016

The present authors' comment, referring to the discussion paper titled "3D-hydrodynamic modelling of flood impacts on a building and indoor flooding processes", is aimed at the comment of anonymous referee #1, published on 19 Feb 2016.

The authors of the manuscript would like to thank the reviewer for the valuable comment. It is addressed, as reported below, to the issues (1) to (4), which are commented by the authors as follows:

(1) No assessment and consideration of on-site measurements or past damages in the case study area and with it no adequate model calibration validation has been accomplished

[Figure]

According to what is known, no previously occurred flood event at the Rio Vallarsa caused relevant damages in the case study area and on the building which was selected for computational modelling. As stated in section 2.2 of the manuscript, a flood event occurred in November 2012 and the Department of Hydraulic Engineering (Autonomous Province of Bolzano) assume that channel geometry can cope with discharges in the range 30-40 m$^3$/s before overbank flooding occurs. The data and information from the 2012-event led to an adaptation of the HQ100- and HQ300-discharge design hydrographs and this was also considered in the present work (section 2.2), which is primarily addressed on this design flood (HQ300) conditions (section 2.3, first paragraph). With the available information the computational model was accurately calibrated by adjusting the surface roughness parameters in the channel. As stated in section 2.3 (first paragraph) of the manuscript and already discussed in the work of Hofer (2014), discharges higher than 30 m$^3$/s exceed the channel capacity in the model which fits well with the available information and expert assessment. In summary, the roughness coefficients are well calibrated in the torrent channel and – since any observation data is not available – set to characteristic values found in literature for the floodplain and the building structure.

Since the aspect of model calibration is also mentioned in the comment of anonymous referee #2, the sections 2.2 and 2.3 of the manuscript will be revised accordingly within the further revision process.

(2) No consideration of uplift processes that potentially (also) damage the considered building

Based on the physics-based vulnerability assessment scheme for buildings exposed to torrential hazards (Figure 2 and introduction-section, both referring to the work of Mazzorana et al. (2014)), the present work means a first step towards this integral assessment concept. It is explicitly addressed on hydrodynamic modelling of building intrusion processes and thereby mainly addressed to analyse the general need and the added value of complex three-dimensional computations compared to con-

ventional two-dimensional flood plain mapping. Any structures in the model (terrain, building walls, etc.) are assumed to be rigid obstacles with a certain surface characteristics. The focus was set primarily mainly on three research questions in the manuscript (section 1, last paragraph): (i) relevance of dynamic clear-water impacts on the building, (ii) delivery of beneficial information for the design of local structural protection measures and (iii) computational possibilities and limits (from a practical rather than a purely scientific perspective) of complex three-dimensional modelling with regard to its application on larger areas (floodplains) with at least a couple of buildings. The authors conclude in the manuscript that an adequate modelling of indoor flooding processes has a rather small influence on the adjacent flow field. Consequently, inundation mapping does not necessarily require a three-dimensional modelling approach. However, for the analysis of local structural protection measures at the building, this modelling approach delivers very valuable information, e.g. (i) points in time and locations of initial indoor flooding, (ii) critical loading conditions, (iii) determination of critical and safe locations inside the building, (iv) information for evacuation planning or (v) efficiency analysis of various options of protection measures. By reflecting the simplifying assumptions of the modelling concept (compared the figure 2 and the contents of the introduction-section) the mentioned modelling benefits and computational effort are discussed in the sections 3.3 and 4.

(3) Missing consequences / benefit for the planning of new buildings

The authors fully agree with the reviewer's note that consequences and benefits for planning new buildings, potentially resulting from the case study analysis at the Rio Vallarsa torrent, are not clearly stated in the conclusions. With regard to the computed impacts and wetting durations on the considered building, it seems not reasonable to transfer computed specific impact loads under design flood conditions to any further objects. The modelling results showed that the computed impacts and the flooding inside the building are significantly influenced by the design flood characteristics (hydrology and if relevant sediment transport processes), the capacity of the torrent channel and

also by the topography of the adjacent floodplain. As stated already for issue (1), the general knowledge of a reasonable application of three-dimensional models for simulation of indoor flooding processes and, further, its computational limits represent the actual added value of the present case study analysis.

Within the further revision process this aspect is discussed in more detail.

(4) Manuscript language and style

The manuscript is found to be of appropriate language quality, even though no detailed English proofreading has been performed by the reviewer. In this regard the manuscript is again carefully checked within the further revision process.

References

Hofer, T.: 3D-numerische Modellierung der Durch- und Umströmung von Infrastrukturobjekten (Gebäuden). Master thesis, Unit of Hydraulic Engineering, University of Innsbruck, 2014 (in German).

Mazzorana, B., Simoni, S., Scherer, C., Gems, B., Fuchs, S., and Keiler, M.: A physical approach on flood risk vulnerability of buildings. Hydrol. Earth Syst. Sci. 18, 3817-3836, 2014. Doi: 10.5194/hess-18-3817-2014